# In Situ Monitoring of a Eutrophicated Pond Revealed Complex Dynamics of Nitrogen and Phosphorus Triggered by Decomposition of Floating-Leaved Macrophytes

Cuiyu Yi, Jiafeng Li, Chenrong Zhang, Fan Pan and Changfang Zhou *

School of Life Sciences, Nanjing University, Nanjing 210023, China; yicuiyu21@163.com (C.Y.);
ljf_robert@sina.com (J.L.); pozalady@163.com (C.Z.); cy7a7cy@163.com (F.P.)
* Correspondence: zcfnju@nju.edu.cn

**Abstract:** To explore the influence of the decomposition of aquatic macrophytes on water quality in eutrophicated aquatic ecosystems and the interacting environmental factors that trigger nitrogen (N) and phosphorus (P) dynamics, a suburban pond with floating-leaved macrophytes (Pond A) as well as another nearby newly dug pond without any obvious aquatic macrophytes (Pond B) were studied. N and P levels together with a series of parameters relating to biomass, water and sediments were monitored during a period of 84 d that covered the entire decomposition process of plants. The results show that the decomposition of aquatic macrophytes can be divided into two phases, with the first phase having a faster decomposition rate and the second phase, a slower one. With the decomposition of biomass, the dissolved oxygen (DO), oxidation-reduction potential (ORP), and pondus hydrogenii (pH) of the water body increased, whereas the permanganate index ($COD_{Mn}$) decreased. Significantly higher levels of total phosphorus in both water and sediment ($TP_W$ and $TP_S$) were detected in Pond A with macrophytes; $TP_W$ increased quickly during the first phase of biomass decomposition but decreased in the second phase, and $TP_S$ remained relatively stable during the first phase but increased slowly in the second phase. Total nitrogen in both water and sediment ($TN_W$ and $TN_S$) was also significantly higher in Pond A but remained relatively stable. A structural equation model revealed that the decomposition of aquatic macrophytes, could, directly and indirectly, influence N and P cycles in an aquatic ecosystem through the regulation of pH and DO. Our study indicate that the decomposition of biomass exerted a greater influence on P than on N. Besides the direct release of P from decaying biomass, which caused a significant increase of P in water body, changes of DO and ORP and the subsequent redox state of the whole system during the process also indirectly affected the deposition and dissolution of P between sediment and water. P was the decisive factor that caused endogenous eutrophication in ponds containing aquatic macrophytes.

**Keywords:** floating-leaved macrophytes; decomposition; nitrogen; phosphorus

## 1. Introduction

Aquatic plants, as an important part of wetland ecosystems, purify the water by absorbing nutrients, heavy metals, and other harmful substances, filtering and intercepting sediments, enriching dissolved oxygen, etc. [1,2]. Hence, aquatic plants have been widely used for the restoration of wetland ecosystems, especially in eutrophicated water bodies, in which excessive nitrogen (N) and phosphorus (P) contribute to the major problems [3]. Meanwhile, increasing attention has been paid to the decomposition process of aquatic plants. In addition to the exogenous N and P that cause eutrophication [4,5], the decomposition of aquatic plants and subsequent release of N, P, and other components from decayed biomass have been treated as an endogenous source of pollutants [6,7].

For the two nutrients, N and P, although they are simultaneously released from the decomposed biomass, they have different dissolution processes, transforming within the

water body and transferring between the water–sediment interface. Meanwhile, more and more aquatic scientists nowadays agree that P is the key factor that controls algal blooms, one of the typical phenomena of eutrophicated freshwater ecosystems [8]. The fact that P can bind with metals such as Fe, Ca, and Al and deposit in the sediment makes it complicated in terms of its biogeochemistry [9].

Furthermore, water quality does not depend only on total amounts of N and P within the wetland ecosystem; there are substantive parameters such as dissolved oxygen level (DO) and oxidation-reduction potential (ORP) that interact with the nutrients and determine the overall water quality. Contrasting opinions on the influence of aquatic plants on water quality during their decomposition process have been reported. Most of the studies report that the decomposition of aquatic plants negatively impacts water quality because of the release of large amounts of N, P, and other nutrients back into the water, as mentioned above [10–12]. However, some other studies also argue that the decomposition of aquatic plants has positive effects on water quality, as the process promotes denitrification in the system [13] and improves the transparency and permanganate index ($COD_{Mn}$), etc. of the water body [14]. Hence, although it is still appropriate to focus on the N and/or P levels as the key factors when evaluating the water quality of eutrophicated aquatic ecosystems, simultaneous interacting parameters should be considered, which can help us to evaluate the detailed dynamics of N and P.

At present, methods to research the decomposition of aquatic plants mainly include indoor simulation [15] and litterbags [16,17]. The former is mainly used to explore the effects of certain quantitative factors on plant decomposition, and most of them are small-scale simulation experiments [18] in which the water volume is normally limited. The latter puts a small amount of plant samples into litterbags and periodically measures the plant-related indicators [19], which have been widely used to study the loss of plant biomass and changes in the nutrient contents [20], and also used to study the ecosystems with relatively long nutrient gradients and large spatial scales [21]. Although studies have declared there is little or no influence from the litterbags themselves [22], it will be expected that an alternative field study without the use of them be designed.

In this study, we designed a set of pond ecosystems in a suburban area of our city. An old, seriously eutrophicated pond (A) fully covered by floating-leaved macrophytes and with deep silt on the pond floor was selected, along which another new pond (B) was dug, which had no obvious aquatic macrophytes nor collectible silt on the floor. A small volume of overflow from Pond A to Pond B may have occurred during rainy seasons in spring and summer when plants grew, whereas they were entirely separated during the dry seasons in autumn and winter when plants decomposed, as little rain or snow fell, and the water level was, at times, low. A third small pond (O) was also dug between A and B, and enough aquatic macrophytes were transplanted inside to remove possible nutrients released from Pond A into Pond B during the rainy season. Therefore, during the season where decomposition of aquatic macrophytes occurred, both ponds A and B could be treated as relatively enclosed systems without much exchange of chemicals with outside except for volatile compounds. Thus, we could simplify our study on the nutrient cycle among plant biomass, water body and sediment inside the pond ecosystem and the consequent influence of vegetation deposition on the water body by comparing the data between ponds A and B.

To reveal the trends of water quality during aquatic macrophyte decomposition after the growing season and the underlying mechanism, an in situ long-term monitoring program was designed in this study. A series of biomass, water, and sediment parameters were detected over a period of 84 days, during which the majority part of the aquatic biomass was decomposed. A structural equation model was also developed to better understand the relationships between the parameters monitored and the driving force(s) of water quality change during plant decomposition.

## 2. Materials and Methods

### 2.1. Study Area

The ponds were located in the north of Dawang Village, Jiangning District, Nanjing City, Jiangsu Province (31°48′ N, 118°46′ E). The climate of the study area was subtropical monsoon with a mean annual precipitation of 1267 mm, which was mainly concentrated between April and August. The monitoring period was from October 2018 to January 2019, and the variation of mean daily air temperature is shown in Figure 1 (the data were collected from the local meteorological monitoring station).

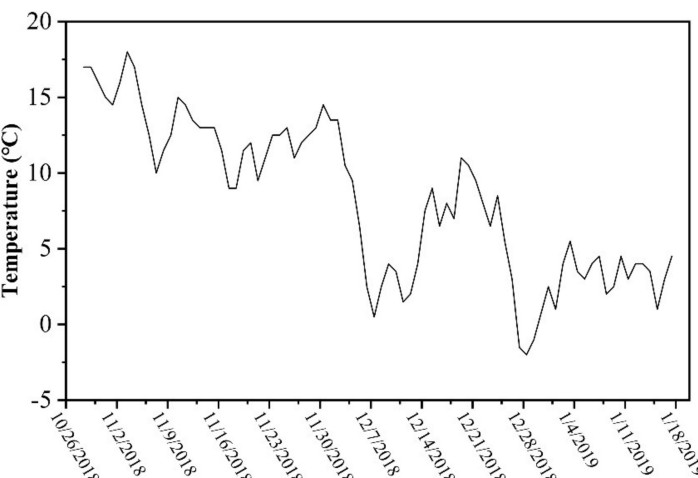

**Figure 1.** Mean daily air temperature of the research area during the monitoring period.

As mentioned above, floating-leaved macrophytes proliferated in Pond A, and the initial coverage rate reached 100% (Figure 2). According to the dominant species, this pond can be divided into 3 patches: Patch a dominated by *Trapa bispinosa* Roxb, Patch c domintated by *Azolla imbircata* (Roxb.) Nakai, and Patch b with a mix of the two species. Since no significant difference in the water quality parameters was found among the 3 patches ($p > 0.05$), we considered Pond A as a water body with homogeneous physicochemical properties, and 5 sampling sites, S1 to S5, were retained. Pond A had a surface area of 3500 m$^2$ during the dry season, with a silt thickness of 0.8 to 1.3 m and water depth of 0.5 to 1.3 m. Approximately 50 m away was the newly dug Pond B, with an area of 2800 m$^2$ and water depth of approximately 1.0 m, in which sampling sites S6 to S10 were located. The additional pond O was located in the middle between ponds A and B, with an area of approximately 400 m$^2$ and depth of 0.5 m.

### 2.2. Sampling and Analyses

Basic properties of floating-leaved macrophytes were measured on 27 October 2018. Fresh samples of *A. imbircata* and *T. bispinosa* were collected and dried in an oven at 105 °C for 30 min, and then at 60 °C to constant weight. The total phosphorus (TP) of the dried samples was determined by the H$_2$SO$_4$-H$_2$O$_2$ desiccation method [23], and the total nitrogen (TN) was determined by potassium the persulfate oxidation absorption photometric method [24]. The total organic carbon content (TOC) was determined by potassium dichromate oxidation spectrophotometry (HJ615-2011).

Field sampling for plant coverage and biomass, water quality parameters, and sediment parameters was conducted every 2 weeks a total of 7 times.

#### 2.2.1. Plant Coverage and Biomass

Five quadrats (1 m × 1 m) were randomly selected from each of the 3 patches to estimate the plant coverage and measure the biomass (fresh weight) of the floating-leaved macrophytes.

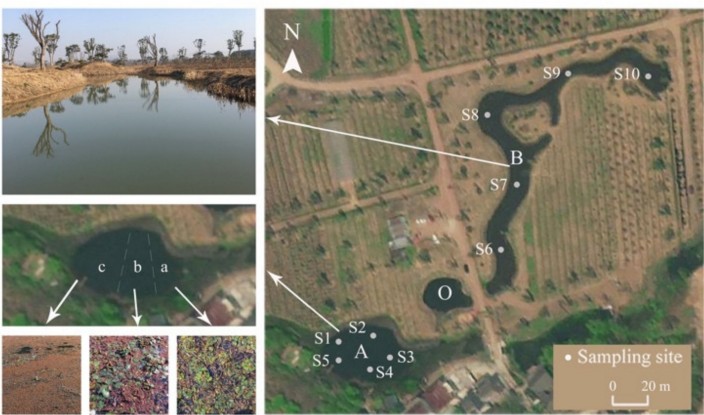

**Figure 2.** Sampling site distribution of water and sediment. A: Pond A, B: Pond B, O: Pond O. a: Patch a dominated by *Trapa bispinosa* Roxb, c: Patch c dominated by *Azolla imbircata* (Roxb.) Nakai, b: Patch b with a mix of the two species. S1–S5: sampling sites of Pond A, S6–S10: sampling sites of Pond B.

### 2.2.2. Water Quality Parameters

Water quality parameters of the 2 ponds were measured on site. DO was measured using a portable dissolved oxygen meter (Leici, JPBJ-608, Shanghai, China). ORP and pondus hydrogenii (pH) were measured using pH/ORP meter (Hanna, HI98160, Woonsocket, RI, USA); each parameter was measured 20 times. The water samples were collected, and total phosphorus ($TP_W$), total nitrogen ($TN_W$), and $COD_{Mn}$ were measured immediately on the day of field sampling. $TP_W$ was determined by the ammonium molybdate spectrophotometer method (GB11893-89), $TN_W$ was determined by potassium persulfate digestion UV spectrophotometer method (HJ636-2016), in which potassium persulfate needed to be recrystallized. $COD_{Mn}$ was determined by the potassium permanganate oxidation method (GB11892-89).

### 2.2.3. Sediment Parameters

Sediment samples were collected at the same sampling sites as the plant and water quality parameters. Samples were air-dried and the basic physical-chemical parameters were measured by Bao's [23] method. The Kjeldahl method was adopted to measure the total nitrogen of sediment ($TN_S$). The total phosphorus of sediment ($TP_S$) was determined by the $HClO_4$-$H_2SO_4$ digestion-Mo-Sb Anti spectrophotometric method.

### 2.3. Statistical Analysis

Statistical analysis was conducted in SPSS 13.0. One-way Analysis of Variance (ANOVA) and post hoc Dunnett's tests was adopted to determine the differences in coverage and biomass of the 3 patches in Pond A. Among different ponds, the water quality parameters and sediment paraments were analyzed by an independent-simple *t*-test, and the difference was considered significant at $p < 0.05$. In Pond A, the correlation between biomass and environmental factors was analyzed by Spearman correlation analysis. Redundancy Analysis (RDA) was conducted in Canoco 4.5, and the structural equation model (SEM) was drawn using Amos. The overall fit degree of the model was based on the normed fit index (NFI), goodness-of-fit index (GFI), comparative fit index (CFI), incremental fit index (IFI), and root mean square error of approximation (RMSEA).

## 3. Results

### 3.1. Coverage and Biomass of Pond A

During floating-leaved macrophyte decomposition, a decreasing trend was observed in the coverage (Figure 3a) and biomass (Figure 3b) in the three patches of Pond A. In the first phase of plant decomposition (0–42 d), the coverage decreased slowly and there was

no significant difference among the three patches ($p > 0.05$), while the biomass decreased rapidly and the rate of biomass loss in Patch a dominated by *A. imbircata* was lower than that in the other two patches. In the second phase of decomposition (42–84 d), the coverage decreased rapidly, and the three patches began to show significant differences ($p < 0.05$). At the end of the monitoring period, the coverage of the Patch c dominated by *A. imbircata* was reduced by 79%, and that of the Patch b with a mix of two species was reduced by 44%, and that of the Patch a dominated by *T. bispinosa* reduced by 24%. During the whole monitoring period, the biomass of *T. bispinosa* decreased by 76%, that of the mix of *A. imbircata* and *T. bispinosa* decreased by 80%, and that of the *A. imbircata* decreased by 69%.

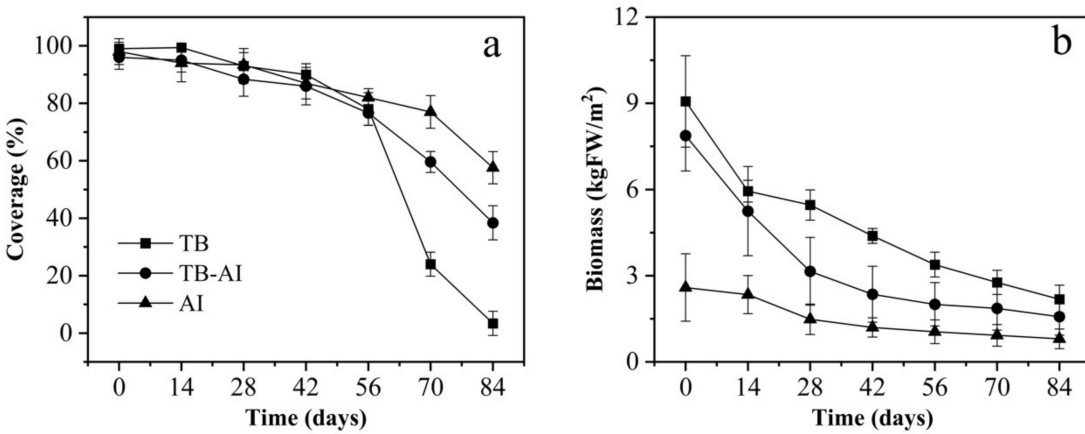

**Figure 3.** Dynamics of coverage (**a**) and biomass of floating-leaved macrophytes (**b**) in Pond A. TB: *T. bispinosa*, TB-AI: *T. bispinosa* and *A. imbircata* mixed, AI: *A. imbircata*. Data are listed as the mean ± standard deviation (SD), $n = 5$.

The initial physicochemical properties of *A. imbircata* and *T. bispinosa* are shown in Table 1. TN and TP in *A. imbircata* were significantly lower than those in *T. bispinosa* ($p < 0.05$), but there was no significant difference in TOC between them ($p > 0.05$). Therefore, TOC/TN and TOC/TP in *A. imbircata* were significantly higher than those in *T. bispinosa* ($p < 0.05$).

**Table 1.** Initial physicochemical properties of floating-leaved macrophytes.

| Parameters | TN/(g/kgDW) | TP/(g/kgDW) | TOC/(g/kgDW) | TOC/TN | TOC/TP |
|---|---|---|---|---|---|
| AI | 22.12 ± 1.74 [b] | 0.75 ± 0.08 [b] | 265.2 ± 36.14 [a] | 12.09 ± 2.08 [a] | 357.03 ± 68.25 [a] |
| TB | 34.25 ± 1.92 [a] | 2.13 ± 0.28 [a] | 240.9 ± 34.21 [a] | 7.06 ± 1.15 [b] | 114.89 ± 23.99 [b] |

AI—*A. imbircata*; TB—*T. bispinosa*. Different lowercase letters [a] and [b] in the table indicated significant difference in the same data index between *A. imbircata* and *T. bispinosa*, $p < 0.05$.

### 3.2. Water Parameters

The DO (Figure 4a) was significantly lower in Pond A than that in Pond B at each sampling time ($p < 0.05$). DO fluctuated between 9.11 mg/L to 10.68 mg/L in Pond B, while it had an overall upward trend in Pond A. During the decomposition of floating-leaved macrophytes, DO in Pond A was relatively low and stable at 0–42 d, while it rapidly increased to 5.30 mg/L at 42–56 d, and then slightly increased to 6.22 mg/L at 56–84 d.

The ORP of Pond B (Figure 4b) was relatively stable, with a mean value of 195 mV. The ORP of Pond A was negative the first three times and positive the last four times and showed an overall upward trend. The ORP of the first six times was lower than that of Pond B, and the difference between the two ponds was significant ($p < 0.05$). Due to the sharp increase in the ORP values in Pond A during 28–42 d, those of the two ponds were almost the same at the end of the monitoring period ($p > 0.05$).

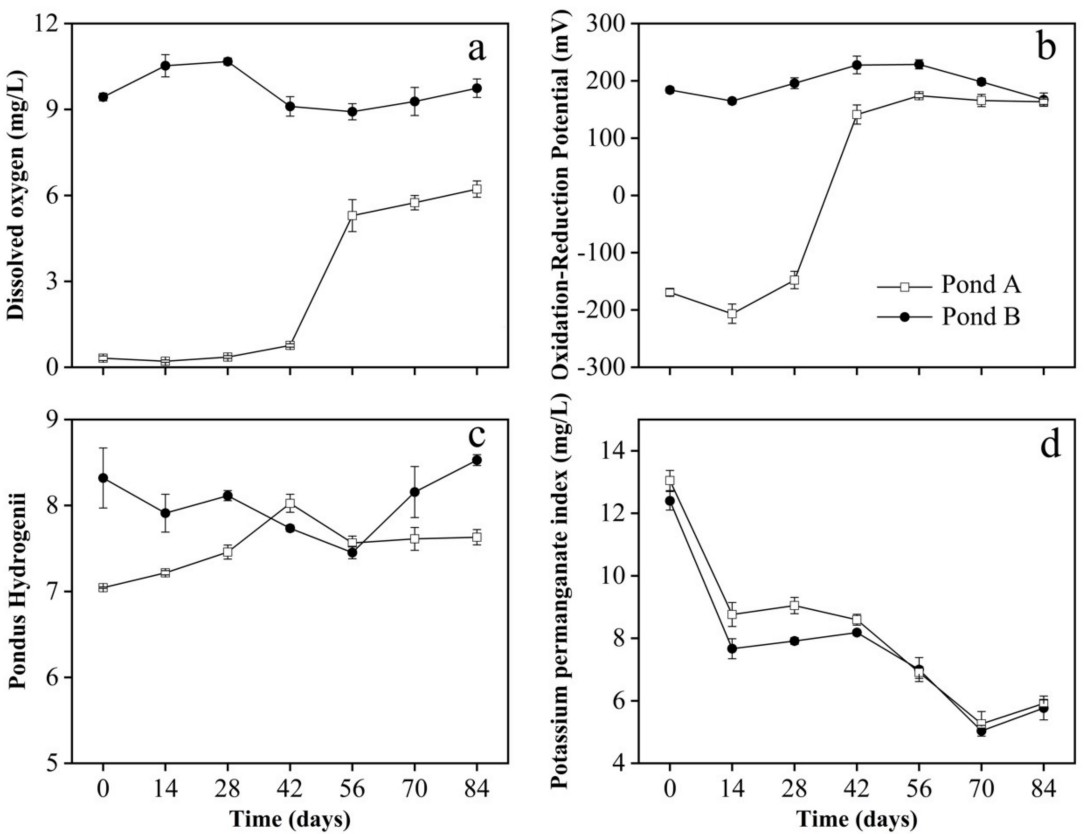

**Figure 4.** Dynamics of dissolve oxygen (DO, (**a**)), oxidation-reduction potential (ORP, (**b**)), pondus hydrogenii (pH, (**c**)) and permanganate index ($COD_{Mn}$, (**d**)) among the two ponds. Data are listed as the mean ± standard deviation (SD). For $COD_{Mn}$, n = 5, for all the other parameters, *n* = 20.

The final pH values of both ponds (Figure 4c) were higher than the initial ones. The pH of Pond A increased first and then decreased, while the pH of Pond B showed the opposite trend. There were significant differences in pH between the two ponds at each sampling time ($p < 0.05$).

The $COD_{Mn}$ in the two ponds (Figure 4d) showed the same downward trend. We speculated that the decomposition of aquatic macrophytes had little effect on $COD_{Mn}$.

$TN_W$ (Figure 5a) and $TP_W$ (Figure 5b) in the two ponds showed different trends. Throughout the decomposition process, $TP_W$ of Pond A was significantly higher than that of Pond B ($p < 0.05$). $TP_W$ increased from 0.108 mg/L to 0.328 mg/L in the first phase of decomposition and decreased rapidly at 28–56 d, before stabilizing at 56–84 d. $TN_W$ in Pond B increased from 0.856 mg/L to 1.409 mg/L, showing an overall upward trend. $TN_W$ in Pond A fluctuated and was always consistently higher than that in Pond B. The difference in $TN_W$ between the two ponds was significant ($p < 0.05$), except for those at the last sampling time.

*3.3. Sediment Parameters*

$TP_S$ (Figure 6a) and $TN_S$ (Figure 6b) in pond A were significantly higher than those in Pond B throughout the monitoring period ($p < 0.05$). The $TP_S$ in Pond B was stable between 0.205–0.254 g/kg, and $TN_S$ increased slowly from 0.26 to 0.62 g/kg. $TP_S$ and $TN_S$ in pond A remained stable during the first phase of plant decomposition, while TPs gradually increased to 0.911 g/kg at the end of the monitoring period, and $TN_S$ fluctuated in the second phase of decomposition.

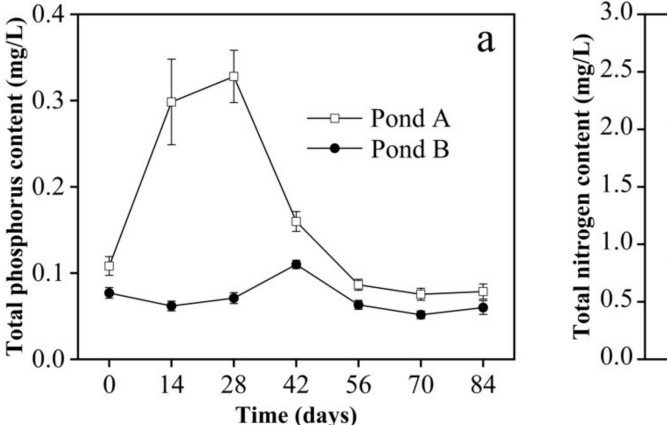
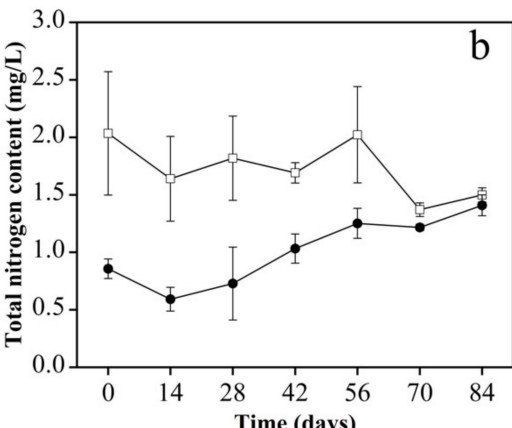

**Figure 5.** Dynamics of total phosphorus (TP$_W$, (**a**)) and total nitrogen (TN$_W$, (**b**)) of water among the two ponds. Data are listed as the mean $\pm$ standard deviation (SD), *n* = 5.

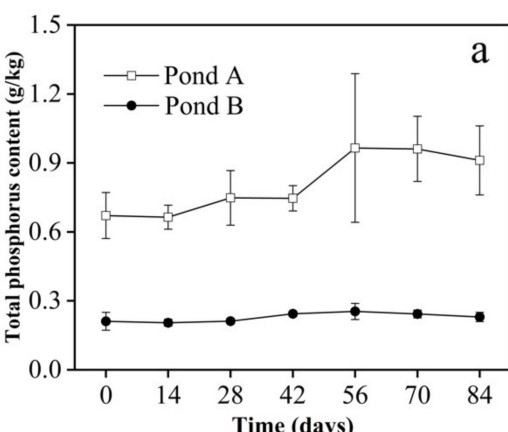
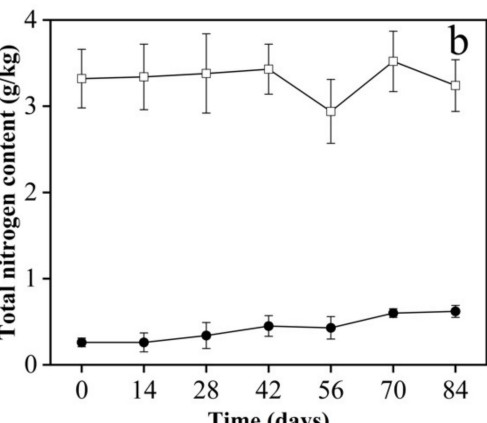

**Figure 6.** Dynamics of total phosphorus (TP$_S$, (**a**)) and total nitrogen (TN$_S$, (**b**)) of sediment among the two ponds. Data are listed as the mean $\pm$ standard deviation (SD), *n* = 5.

### 3.4. Correlations between Floating-Leaved Macrophytes and Environmental Parameters

The first axis, which explained 74.95% of the variation in environmental factors, was primarily associated with *T. bispinosa*, *A. imbircata*, a mix of *T. bispinosa* and *A. imbircata* mixed (Figure 7). Redundancy analysis (RDA) helped determine the independent contribution of the three types of hydrophytes to the water and sediment parameters. The results showed that *T. bispinosa* contributed the most to the environmental factors ($F$ = 72.9, $p$ = 0.002), describing 68.8% of the variation. *A. imbircata* followed ($F$ = 4.6, $p$ = 0.03), describing 3.9% of the variation. The contribution of a mix of *A. imbircata* and *T. bispinosa* to environmental factors was the least ($F$ = 2.7, $p$ = 0.12), which only described 2.2% of the variation.

With the results of RDA analysis, we could focus on the biomass of *T. bispinosa*, and further quantified the correlation between the biomass and the environmental factors with Spearman correlation analysis (Table 2). It was found that biomass of *T. bispinosa* was negatively correlated with DO, ORP, pH, and TP$_S$ ($p$ < 0.05) but positively correlated with TP$_W$, TN$_W$ and COD$_{Mn}$ ($p$ < 0.05). Among the environmental factors, DO was positively correlated with ORP, pH, and TP$_S$ ($p$ < 0.05) and negatively correlated with TP$_W$ and COD$_{Mn}$ ($p$ < 0.05). ORP was positively correlated with TP$_S$ and pH ($p$ < 0.05) and negatively correlated with TP$_W$ and COD$_{Mn}$ ($p$ < 0.05). TP$_W$ was negatively correlated with TP$_S$, pH, and COD$_{Mn}$ ($p$ < 0.05). TN$_W$ was positively correlated with COD$_{Mn}$ ($p$ < 0.05). pH was negatively correlated with COD$_{Mn}$ ($p$ < 0.05). No significant correlation was found between TN$_S$ and any measured factors ($p$ > 0.05) (Table 2).

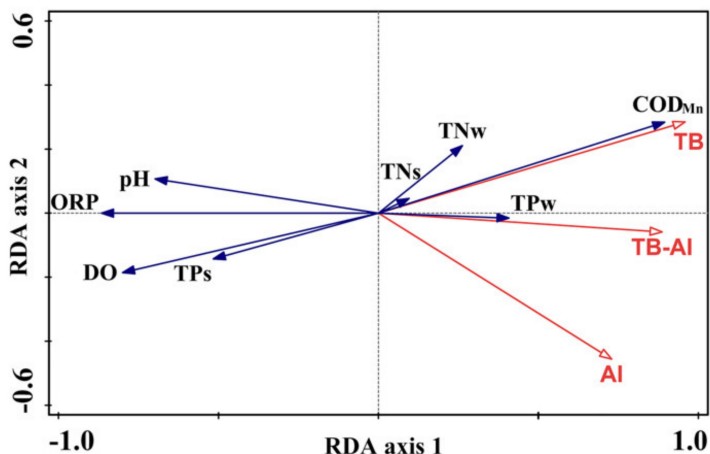

**Figure 7.** Ordination biplot of redundancy analysis (RDA) displaying the effects of biomass on environmental variables. TB—*T. bispinosa*; TB-AI—*T. bispinosa* and *A. imbircata* mixed; AI—*A. imbircata*; DO—dissolved oxygen; ORP—oxidation-reduction potential; $TP_W$—total phosphorus content of water; $TP_S$—total phosphorus content of sediment; $TN_W$—total nitrogen content of water; $TN_S$—total nitrogen content of sediment; pH—pondus hydrogenii; $COD_{Mn}$—permanganate index.

**Table 2.** Spearman correlation coefficients among biomass of floating-leaved macrophytes and environmental factors.

| Parameters | Bio | DO | ORP | $TP_W$ | $TP_S$ | $TN_W$ | $TN_S$ | pH | $COD_{Mn}$ |
|---|---|---|---|---|---|---|---|---|---|
| Bio | 1 | | | | | | | | |
| DO | −0.937 ** | 1 | | | | | | | |
| ORP | −0.777 ** | 0.833 ** | 1 | | | | | | |
| $TP_W$ | 0.726 ** | −0.796 ** | −0.710 ** | 1 | | | | | |
| $TP_S$ | −0.694 ** | 0.756 ** | 0.603 ** | −0.609 ** | 1 | | | | |
| $TN_W$ | 0.386 * | −0.376 * | −0.131 | 0.289 | −0.490 ** | 1 | | | |
| $TN_S$ | 0.078 | −0.080 | −0.153 | 0.073 | −0.134 | −0.055 | 1 | | |
| pH | −0.778 ** | 0.741 ** | 0.593 ** | −0.395 * | 0.527 ** | −0.262 | 0.063 | 1 | |
| $COD_{Mn}$ | 0.895 ** | −0.838 ** | −0.770 ** | −0.717 ** | −0.662 ** | 0.559 ** | 0.042 | −0.662 ** | 1 |

**—Significance at $p < 0.01$, *—significance at $p < 0.05$. Bio—biomass of *T. bispinosa*; ORP—oxidation-reduction potential; $TP_W$—total phosphorus content of water; $TP_S$—total phosphorus content of sediment; $TN_W$—total nitrogen content of water; $TN_S$—total nitrogen content of sediment; pH—pondus hydrogenii.

### 3.5. Structural Equation Model

Here, we also selected the biomass of *T. bispinosa* to develop the SEM model. DO and $COD_{Mn}$ were eliminated because the biomass of *T. bispinosa* had multiple linearity with DO and $COD_{Mn}$. We selected the following parameters related to the decomposition of floating-leaved macrophytes: ORP, pH, $TP_W$, $TP_S$, $TN_W$ and $TP_S$. Based on the hypothesis of aquatic macrophytes decomposition, we constructed the SEM ($\chi^2 = 8.154$, $p = 0.319$, df = 7). The maximum likelihood method was adopted to estimate the model. The fitting results (Table 3) showed that the model has a reasonable fit, which reflects the most probable contribution of aquatic macrophytes decomposition to water quality and sediment parameters.

**Table 3.** Global fitting index of the structural equation model.

| Fitting Index | $\chi^2$/df | GFI | NFI | CFI | IFI | RMSEA |
|---|---|---|---|---|---|---|
| Standard value | <3 | >0.9 | >0.9 | >0.9 | >0.9 | <0.08 |
| Modified fitting value | 1.165 | 0.940 | 0.936 | 0.989 | 0.990 | 0.070 |

The model (Figure 8) showed that the biomass of *T. bispinosa* was inversely proportional to ORP, $TP_S$, and pH, but directly proportional to $TP_W$ and $TN_W$, which was in accordance with the Spearman correlation analysis. The parameters included in the model

explained 76% of the variation in ORP, 44% in $TP_S$, 42% in pH, 17% in $TN_W$, and 2% in $TN_S$. A total of 55% of the $TP_W$ variation was explained by direct and indirect effects. In addition to the direct effect on N and P levels of both water and sediment, the decomposition biomass also had an indirect effect through pH. $TP_W$ and $TP_S$ were also indirectly affected through ORP.

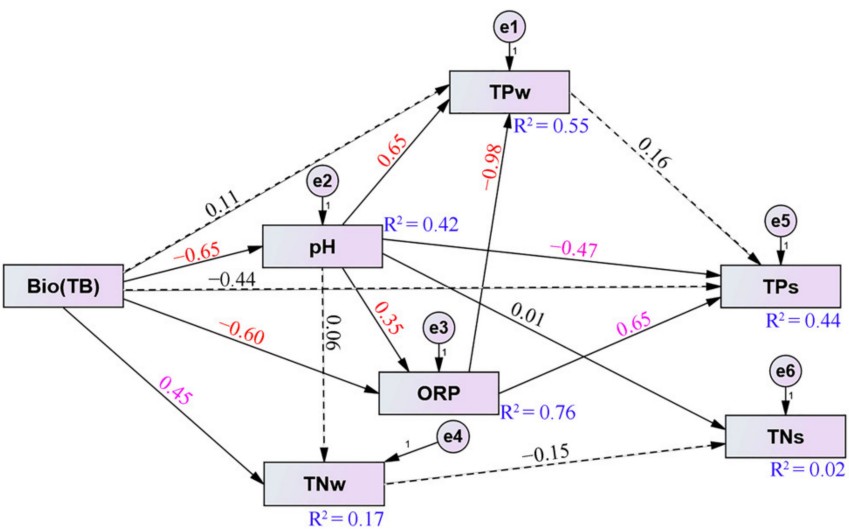

**Figure 8.** A structural equation model (SEM) of the causal relationships among biomass, ORP, pH, $TP_W$, $TP_S$, $TN_W$, and $TN_S$ dynamics in the water column. The black numbers above dashed arrows are the path coefficient with significance at $p > 0.05$. The red numbers above solid arrows are the path coefficient with significance at $p < 0.01$. The pink numbers above solid arrows are the path coefficient with significance at $p < 0.05$. The $R^2$ values represent the proportion of variance explained for each endogenous variable. e1–e6: residual term. Bio(TB)—biomass of *T. bispinosa*; ORP—oxidation-reduction potential; $TP_W$—total phosphorus content of water; $TP_S$—total phosphorus content of sediment; $TN_W$—total nitrogen content of water; $TN_S$—total nitrogen content of sediment; pH—pondus hydrogenii.

## 4. Discussion

### 4.1. Decomposition Process of Floating-Leaved Macrophytes

In this study, we have used two ponds to conduct our in situ monitoring. As for the difference between the two ponds, Pond A was an old one fully covered with floating-leaved macrophytes and thick silt, while Pond B was newly dug, in which initial succession of aquatic macrophytes and silt accumulation had not started. Hence, we could hypothesize that the fluctuations of N and P levels in the waterbody of Ponds A during our monitoring period were mainly caused by the decomposition of biomass. One would argue that the sediment might exert strong effects on the waterbody through physicochemical deposition and redissolution of N and P on the water–sediment interface. We thought that it was also influenced by environmental factors such as DO, which were eventually triggered by the decomposition process.

Our study revealed two obvious phases in the decomposition of floating-leaved macrophytes, with the first phase being faster and the second slower, which is in agreement with former reports of wetland plants decomposition with litterbags [25,26], indicating our initial hypothesis that treating the ponds in drying season without much water exchange as relatively enclosed ecosystems was practical, and the litterbags could be skipped in similar research. In the first phase, the leaching of water-soluble substances was the primary mechanism for the loss of litter mass [27], which released phenols, sugars, and elements such as N, P, and potassium into water. In the second phase, microbial decomposition and transformation of hard-to-decompose substances (such as cellulose) were mainly carried out in water and sediment [26]. The fragmentation of aquatic macrophytes residues was

also affected by micro-, meio- and macroinvertebrates [28]. Abiotic factors such as temperature, DO, and pH may further affect the decomposition of aquatic macrophytes [29,30]. Consistent with the reports of Enriquez et al. [31] and Chimney et al. [32], decomposition rates were negatively correlated with litter C:N and C:P molar ratios and positively correlated with N and P contents. The decomposition rate in our study was fast in Patch a dominated by *T. bispinosa*, while relatively slow in Patch c dominated by *A. imbircata*. This was due to the higher initial TN, TP, and lower C:N and C:P molar ratios in *T. bispinosa* compared to those in *A. imbircata*.

### 4.2. Effect of Floating-Leaved Plant Decomposition on Nitrogen and Phosphorus

It is well known that the decomposition of plant biomass increases the amount of organic matter in the environment [33], and it is also considered as the carbon source of constructed wetlands [34]. The promotion of microbial activity along with the decomposition of aquatic macrophytes was also reported [35]. Therefore, it was reasonable to deduce that the decomposition of floating-leaved macrophytes released a large amount of dissolved organic matter, among which were organic N and P compounds, and this constituted the direct influence of biomass deposition on water N and P levels.

However, from our long-term monitoring results, no linear correlation between N levels ($TN_W$ or $TN_S$) or P levels ($TP_W$ or $TP_S$) and biomass was observed at our research site. Reduction in biomass alone could not explain the variation in N and P levels. Therefore, we built the structural equation model to better understand the relationships between the parameters and the driving force(s) of the N and P dynamics during plant decomposition. The model showed that besides the direct effect of aquatic macrophytes decomposition, regulation of DO and pH could indirectly influence N and P cycles in Pond A.

In our study, significant changes in DO and ORP were detected in relation to biomass decomposition, which we thought, through regulation of the redox state of the water body, indirectly affected the $TP_W$. P can bind to metal ions at the water–sediment interface of aquatic ecosystems, forming Al-bound P (Al-P), Fe-bound P (Fe-P), and Ca-bound P (Ca-P) [36]. Among those inorganic phosphorus compounds, Fe-P is the most unstable. When ORP is low, the water–sediment interface is in the reduced state [37,38], and reductive dissolution of phosphorus–bearing iron (oxyhydr)oxides is regarded as a primary mechanism responsible for the mobilization of phosphorus in sediments [39,40], as $Fe^{3+}$ is transformed into $Fe^{2+}$, and P is released to the water body, which increases the $TP_W$. Therefore, with low DO and ORP in the first deposition phase, we assumed part of the P from sediment in Pond A was also redissolved into the water body, which further increased $TP_W$. With the recovery of DO and ORP in the second phase, the water–sediment system turned into the oxidize state, and P in the water was adsorbed by colloidal $Fe(OH)_3$ and precipitated or combined with $Fe^{3+}$ to form $FePO_4$, then deposited in the silt along the bottom [41], thus $TP_W$ decreased and $TP_S$ increased.

The water pH levels in Ponds A and B were both between 7 and 9, and the overall pH level in Pond A was generally lower than that in Pond B. We assumed that the plant decomposition and the release of organic acids decreased the pH level in Pond A. However, with the data we detected on-site, it was a little difficult to explain the influence of pH on P levels. The SEM model indicated a positive relationship between water pH and $TP_W$ but a negative relationship between water pH and $TP_S$, whereas reversed relationships were observed by RDA and Spearman correlation analysis. With the results from Jin et al. [42], P release was the lowest in neutral conditions, and an increase in pH could free P from its binding to ferric complexes due to the competition between hydroxyl ions and the bound P ions [42,43]. With our understanding, we also suggested that the SEM results could be more reasonable, as increased pH in water promoted the release of P from sediment into the water. The opposite results from RDA and Spearman correlation analysis might be reflections of other environment factors on P. As was mentioned by Chen et al. [44], besides DO and pH, temperature, nitrate, organic matter in the environment as well as the activity of microorganism, etc., could all influence P migration and transformation

across the water–sediment interface. In addition, we only monitored the water pH level, and that of the sediment might be different. Further monitoring of pH level in sediment was recommended in future study. Nevertheless, the contradictory results obtained in this study serve as a reminder of the importance of method selection when analyzing field data.

The N levels in water and sediment of Pond A were both consistently higher than those in Pond B. Unlike P, N in Pond A remained relatively stable during the plant decomposition process. We suggested that the release of N from decayed biomass continued, whereas N had experienced fast nitrogen mineralization, nitrification, denitrification and anammox [45]. As has been suggested, dissolved organic carbon released by plant decomposition enhanced the activity of denitrifying microorganisms [46]. Volatilization of N as $N_2$, $N_2O$ and $NH_3$ in Pond A might balance the N newly released from biomass.

From our research of the N and P dynamics during aquatic macrophyte decomposition, it was difficult or improper to simply conclude if the decomposition process had positive or negative effects on water quality. As for the conflicting opinions raised by different researchers, we thought these were mainly due to the parameters chosen to represent the water quality. Nevertheless, the joint use of multiple quantitative methods of RDA, Spearman correlation analysis, and SEM offers a better explanation of the complex relationships among the parameters in the aquatic ecosystem.

Generally, the decomposition of biomass better explained P levels than N levels in the water body of the ponds we studied, indicating that P was the decisive factor that caused the endogenous eutrophication of ponds during plant decomposition. In Schindler's [8] research, no evidence was found that proved eutrophication can be managed by controlling N inputs, while there was much evidence that controlling P inputs allows management at the whole–lake scale. Therefore, in order to control the outbreak of water eutrophication, it is necessary to remove the floating-leaved macrophytes from the ecosystem before their decomposition. For those eutrophic ponds with thick layers of silt in the sediment, dredging was also recommended.

## 5. Conclusions

The decomposition of floating-leaved macrophytes in our research site could be divided into two phases, with the first phase being faster and the latter slower. The decomposition of biomass exhibited a direct effect on the P level in the water body, while through regulation of DO, ORP, and pH, it also exerted certain indirect effect on the P transformation and transference between the water–sediment interface, which further affected the P level in the water. Although significantly higher $TN_W$ and $TN_S$ were found in Pond A than in Pond B, the apparent N levels fluctuated little during the plant decomposition process, indicating an efficient turning over the rate of N by the pond ecosystem. Therefore, it was the P that played a key role in the endogenous eutrophication of ponds during plant decomposition.

In future work, a continuous long-term monitoring program on the N and P cycles within the two ponds would be of merit. Especially when plenty of algae and aquatic macrophytes begin to settle in Pond B, we can treat it as the initial state of a succession of pond ecosystems. By carefully controlling the exogenous input of N and P into the pond, we can track the dynamics of nutrient cycles alone with the ecosystem succession. Future work may also focus on the P in sediments and its transformation and transference on the water-sediment interface. Attention may also be paid to other environmental issues that interact with the P cycles in the aquatic ecosystems, like the ambient atmospheric exposure to litter decomposition, hydrological tracers, and the vegetation types (for example, emergent and submerged macrophytes), etc. In this study, we only discussed how the P dynamics were triggered by plant decomposition. The P cycle in eutrophicated aquatic ecosystems remains to be explored.

**Author Contributions:** Conceptualization, C.Y. and C.Z. (Changfang Zhou); investigation, C.Y., C.Z. (Chenrong Zhang), J.L., and F.P.; data curation, C.Y.; writing—original draft preparation, C.Y., J.L.; writing—review and editing, C.Z. (Changfang Zhou). All authors have read and agreed to the published version of the manuscript.

**Funding:** This research was funded by National Major Science and Technology Projects of China (2017ZX07602), "San-Xin" Forestry Project of Jiangsu Province (LYSX[2015]16), Science and Technology Plan of Nanjing (201805019), Jiangsu Independent Innovation Funding for Agricultural Science and Technology (CX(20)2018).

**Institutional Review Board Statement:** Not applicable.

**Informed Consent Statement:** Not applicable.

**Data Availability Statement:** The data that support the findings of this study are available from the corresponding author, upon reasonable request.

**Conflicts of Interest:** The authors declare no conflict of interest.

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
