# Peer review of "In Situ Monitoring of a Eutrophicated Pond Revealed Complex Dynamics of Nitrogen and Phosphorus Triggered by Decomposition of Floating-Leaved Macrophytes"

_water, doi:10.3390/w13131751_

Round 1
Reviewer 1 Report
The ms by Yi et al. describes an approach of in-situ monitoring of floating-leaved macrophyte decomposition in a eutrophic lake. It claims to provide the “real influence of aquatic macrophytes decomposition on water quality” – yet I consider the study is far from providing clear answers to this.
My main concerns with the manuscript are:
- It is a pseudoreplication as only one lake and a macrophyte-free lake were considered. See Hurlbert 1984 for advice on the issue of pseudoreplication.
- The manuscript is making statements that are simply not true. L84-85 they say a shortage of litterbag use is effects on macroinvertebrates and microorganisms, but many studies have shown that different mesh sizes overcome this problem and provide conclusive results. It is also not a problem that there is no or little exchange of the material inside with outside, as you need to quantify the loss in mass at some point. There are many more of such statements.
- The control pond is not a control pond, as the two ponds cannot be considered as two equal units having only one difference (presence of macrophytes). The only way to follow the processes is by setting up controlled laboratory or mesocosms experiments with sufficient replicates.
- The main findings as stated in the conclusions are a ‘so what?’ – the two-phased pattern of decomposition of plant material is very well known, and nutrient release from decaying macrophytes has also been addressed by many more studies. The authors ignore a wide range of publications on this issue.
- The manuscript is a descriptive study, a case study with extremely limited relevance to enhance the understanding of nutrient dynamics following macrophyte decomposition.
Author Response
Thanks a lot for your comments and suggestions. We have been trying hard to improve our manuscript regarding to the comments of all the three reviewers, and we have also got MDPI to help edit the language. Please see the attachment for the point-by-point responses of your comments.

Reviewer 2 Report
The study deals with the decomposition of aquatic macrophytes and the impacts on N and P dynamics in eutrophicated aquatic ecosystems. The manuscript in my opinion contains valuable information. However, additions and few scientific errors need to be addressed in order increase the validity of the information and help the potential readers. Therefore a minor revision of the manuscript is suggested before publication.
Specific comments:
Split the first paragraph of the abstract.
L46-48. Repeated sentence in line 39.
L70-72 and 84-86. Add references.
In the discussion section I would suggest a paragraph discussing the bias and assumptions of the study, focusing on the use of the pond B (without macrophytes, different history-management etc.) for comparison with pond A.
Conclusions should be improved and focus on the main findings. Moreover, future work should be suggested.
Author Response
Thanks a lot for your comments and suggestions, please see the attachment for point-by-point responses. As another reviewer recommended, we have got MDPI to help edit the language as well.

Reviewer 3 Report
General comment:
Paper summarizes the researches that have been done in a particular area. The body of the paper is well prepared as an effect of loading rate and planting on treatment removal of nitrogen and phosphorus. Conclusion and future directions are correct. The literature cited is appropriate. Introduction, generally, does not introduce much new information or new results, but rather synthesize a larger body of work, providing a new perspective of reducing phosphorus eutrophication, N and P cycles in an aquatic ecosystem as an effect of aquatic plant decomposition
Specific remarks:
The paper makes a good effort in trying to use ecosystem-level studies of terrestrial carbon to reveal contrasting effects of varying environmental conditions on the release of the continuous effects of decaying organic matter on water quality. Please try to address the followings issue, as a coherent group:
1. In Abstract you have to integrate, interpret and expound conclusions. The reader should to really understand the subtle relevance of a particular result or conclusion. You should provide important background information in a study on the effect of the overlying water environment on the nitrogen and phosphorus release of sediment, nutrient release dynamics associated with native and invasive leaf litter decomposition.
2. The second problem is related to the applied value of the commentary, as presented in paragraph 3.4. Correlations between floating-leaved macrophytes and environmental parameters as a key approach to evaluate water through macrophytes analysis. Because the paper starts with a clear applied focus, one could expect that the ‘approach’ mentioned will turn out to be useful. What can be learned from the modelling of different measures for assessing the temporal dynamics of aquatic and terrestrial litter decomposition in terms of methods, applications, problems and solutions?
3. You must expound on a review on litter decomposition and the research progress of aquatic macrophytes. Just reporting the results and conclusions of other studies is important, but, can apparent implication be resolved through a new outlook or interpretation? What do the results from your paper mean? Is your issue relevant, or is it trivial? Is your topic redundant? Has it already been answered? In conclusion, you ought to show major research directions in the field and summarize the major aspect of your work, especially temporal dynamics of phosphorus during aquatic and terrestrial litter decomposition and wetland plant litter decomposition occurring during the freeze season under disparate flooded conditions. Decomposition of emergent aquatic plant litter under different conditions and the influence on water quality must be restated. Please explain how to prepare data to evaluate floating-leaved macrophytes and environmental parameters
4. Conclusions of separate investigations must be combined into a cohesive way. The characteristics of the composition of various phosphorus forms in sediments should be contrasted and compared. The main merit of this paper is an attempt to present complex management issues in worldwide nutrient usage and its impacts on the ecosystem, bringing together important topics of ‘management’ for biological contamination in the marshes and rivers. Thanks to such simplicity, the message is easy to grasp by wide audiences.
Empirical issue:
Which model or patterns in decomposition rates among photosynthetic organisms: the importance of detritus C: N:P content do you recommend?
Although you are not conducting an experiment in the physical sense, you should consider the evaluation of the effects of the decomposition process of aquatic plants as a source of internal lake phosphorus loadings. You are going to read a body of information and provide a new outlook on a topic.
No attention is paid to that the sediment phosphorus speciation and mobility under dynamic redox condition in accordance with in situ decompositions of submerged plants on water quality. Adapting the research on the application of oxidation-reduction potential has been said to correspond to the decomposition of macrophyte litter and decomposition of emergent macrophytes affecting water quality under carbon and nitrogen.
This correspondence appears a little fuzzy and I think it requires some more clarification. For example, how do we view an organic matter decomposition, which may be independent, as suggested by the efficiency of different monitoring units in representing litter decomposition in ecosystem metabolism? Can you review the disappearance of leaf litter under different woodland conditions as tools to assess the functional integrity of streams and rivers-a systematic?
Responses of decomposition rate and nutrient release of floating-leaved and submerged aquatic macrophytes to on microbial decomposers and leaf and eutrophication and harmful algal blooms. Decomposition characteristics of aquatic macrophytes and their potential application as carbon resource in constructed wetland must be added.
Constructive feedback:
The critical components involved in this technique are:
a) the paper does not clearly indicate that it is reporting preliminary results towards the decomposition of aquatic pioneer vegetation in newly constructed wetlands. Unfortunately, the authors do not outline such lessons in any representative way and instead seem to be mostly repeating what others have already said in research papers. Misleading are constructed such as the early-stage litter decomposition and its influencing factors in the wetland of the biomass production and litter decomposition.
b) Contribution of ambient atmospheric exposure to litter decomposition in aquatic environment and determination of eutrophication and its ecosystem response, water quality improvement should be the use of aquatic macrophytes in water-pollution control. Physico-chemical characteristics affect the spatial distribution of effects of aquatic plants during their decay and decomposition on water quality in a eutrophic shallow lake, nutrient conditions and effects of plant growth form and water substrates on the decomposition of submerged litter. As mentioned introduction effects of oxygen availability and temperature as driving forces for the decomposition of aquatic macrophytes are important. Your RDA may help with interpretation. Why did you not applied it? A small example with a figure will help better explain this. Try to mention also the types of hydrological tracers for assessing transport and the effects of litter quality and living plants on the home-field advantage of aquatic macrophyte decomposition. Is the organic decomposition of macrophytes in a shallow pond/ lake governed by water level and water dynamics? Try to prove by statistical analysis.
Summary:
The paper presents an analysis of methods applied to assess nutrients release and greenhouse gas emission during decomposition in a sediment-water system. The paper, primary, describes the role of biogeochemical cycling of nitrogen and phosphorus in pond/lake and the role of microorganisms in the conversion of nitrogen compounds. The investigation of literature employed a rigorously controlled modelling of biogeochemical cycling of nitrogen in lakes and the role of microorganisms in the conversion of nutrient compounds.
Author Response
Thanks a lot for the comments and suggestions from reviewer 3. We have been trying hard to improve the manuscript as suggested, and we have also got MDPI to help edit the language.Please see the attachment for our point-by-point responses.

Round 2
Reviewer 1 Report
My main concern with the study is the pseudoreplication and the lack of novelty of the study. The revised version has improved in the presentation, specifically English language.
However, the main concerns cannot be overcome by a simple revision. Another study design is necessary.
Author Response
Thank you very much for your kindness in helping us with our manuscript (ID: water-1225015), and we apologize for haven't being able to solve all the problems.

Reviewer 3 Report
Thank you for your improvement.
Author Response
Thanks a lot for accepting our responses, and we do appreciate your every comment and suggestion in improving our manuscript.